# Concurrent Targeting of Expressive Vocabulary and Speech Comprehensibility in Pre-Schoolers with Developmental Language Disorder and Phonological Speech Sound Disorder Features: A Survey of UK Practice

**DOI:** 10.3390/children12111568

**Published:** 2025-11-18

**Authors:** Lucy Rodgers, Nicola Botting, Ros Herman

**Affiliations:** 1Department of Allied Health, School of Health and Medical Sciences, City St Georges, University of London, Northampton Square, London EC1V 0HB, UK; nicola.botting.1@citystgeorges.ac.uk (N.B.); r.c.herman@citystgeorges.ac.uk (R.H.); 2Children’s Speech and Language Therapy, Sussex Community NHS Foundation Trust, Elm Grove, Brighton BN2 3EW, UK

**Keywords:** speech sound disorder, developmental language disorder, intervention

## Abstract

**Highlights:**

**What are the main findings?**

**What are the implications of the main findings?**

**Abstract:**

Background/objectives: Speech sound disorder (SSD) and developmental language disorder (DLD) are common childhood disorders of communication that can also co-occur. This study investigated the reported content, format and delivery of UK speech and language therapists’ (SaLTs) practice when delivering intervention for pre-school children with co-occurring SSD/DLD features when expressive vocabulary and speech comprehensibility are prioritised areas. The findings can be used to inform the development of future interventions and enable reflection on current practice. Methods: A quantitative online survey via Qualtrics enabled the statistical analysis of intervention components from SaLTs from across the UK. The survey questions were based on prior research and the input of an expert steering group. The data were analysed through descriptive statistics. Results: There were 108 full responses from across the UK. For both target areas, the responses highlighted a preference for functional intervention targets, holistic delivery of intervention techniques in different environments, and incorporation of techniques into a variety of activities. Most respondents (97.5%) reported that they would also target phonological awareness (PA), with syllable segmentation being the most commonly reported PA target area for inclusion. Overall, 82.4% of respondents said they would consider dosage when providing their intervention. Conclusions: The findings highlight similarities in UK SaLT practice when targeting aspects of both speech (comprehensibility) and language (expressive vocabulary) concurrently, and an emphasis on functional communication, in addition to being guided by developmental norms. The implications for clinical practice and the development of future interventions are discussed.

## 1. Introduction

In the pre-school years, a profile of co-occurring developmental language disorder (DLD) and speech sound disorder (SSD) features is associated with negative long-term outcomes [1,2]. Yet, knowledge of how speech and language therapists (SaLTs) might best support children within this crucial period of their development is in its infancy [3]. This study explores the clinical decision making reported by UK SaLTs within an online survey, asking them how they would provide intervention for a subgroup of pre-school children with co-occurring features of DLD and SSD, where expressive vocabulary and speech comprehensibility are the prioritised areas of need. As well as informing the development of new interventions for this group, the findings shed light on how current clinical practice is characterised, enabling the analysis of and reflection on alignment with the wider evidence base. This is the first step towards maximising intervention effectiveness and improving outcomes for this vulnerable group.

### 1.1. Developmental Language Disorder (DLD) and Speech Sound Disorder (SSD)

Developmental language disorder (DLD) is characterised by difficulties in understanding and/or using spoken language, where there is no known cause or associated neurodevelopmental difference, such as autism [4]. The prevalence of children with features of DLD at 4–5 years is approximately 7.58% [5]. DLD can affect language production (e.g., using new words and combining words together to make sentences) and comprehension, and has lasting implications for emotional wellbeing and education/literacy [4]. In contrast, SSD is an umbrella term to describe difficulties in the production of speech sounds. SSDs are related to, but distinct from, DLD [6]. This is because a child may present with an SSD without meeting the criteria for DLD, and vice versa [6]. Both SSD and DLD are complex, multi-faceted disorders, with various combinations of differing underlying cognitive difficulties contributing to a child’s unique linguistic profile [7]. Co-occurring features of DLD have been evidenced in as many as 40.8% of 4-year-olds with SSD [8]. Acknowledgement of co-occurring SSD/DLD and how it may present in young children is important; a combined SSD/DLD profile in pre-schoolers is associated with negative long-term outcomes relating to literacy [1], quality of life [2], and persisting speech and language needs [2,9]. Therefore, timely and effective intervention for these vulnerable children is needed.

### 1.2. Interventions for Co-Occurring Features of SSD and DLD

Co-occurring features of DLD/SSD may present throughout childhood; however, we have highlighted that known associations with negative longer-term outcomes are evident from the pre-school years, which are a highly important period within brain development [10]. Therefore, interventions need to be available to pre-school children with co-occurring features. In the pre-school years, speech and language therapy (SLT) can be successful in reducing, or even resolving, early features of DLD or SSD [9,11]. Within this article, the term ‘co-occurring features’ refers to children who have overlapping features of the typical characteristics of SSD and DLD. Often, pre-school interventions targeting features of DLD *or* SSD involve similar components, such as supporting parents and/or embedding intervention techniques into naturalistic activities and routines [12]. To date, these similarities have rarely been combined to target aspects of speech and language concurrently. Notable exceptions include studies that specifically focus on combining phonology and morphosyntactic elements of language [13,14]. These studies provide interesting insights, for example, that it may be beneficial to target morphosyntax first and then phonology within intervention [13]. However, it should be noted that both DLD and SSD are heterogeneous, and that findings relating to one target area (i.e., morphosyntax) may not be generalisable to other target areas, such as vocabulary. The sparsity of research on combined interventions with diverse SSD/DLD target areas is somewhat surprising, given the shared linguistic (phonological) underpinnings of DLD/SSD. These shared underpinnings are particularly notable for children with phonological SSDs, which are characterised by rule-based speech production errors, where the child has difficulties using the appropriate sound contrasts (phonemes) when forming words [15]. Phonology is one area of language that may also be impacted in DLD, as demonstrated in Figure 1 below. Such shared difficulties within phonology may relate to the child’s underlying phonological representations [16,17], that is, their stored knowledge of different speech sounds they use in their vocabulary [18].

### 1.3. Expressive Vocabulary and Speech Comprehensibility

In their seminal ‘Child Talk’ study, Roulstone et al. [19] found that in the absence of accessible and implementable published interventions, SaLTs adapt pre-existing interventions for children with non-co-occurring profiles. The findings from their predominantly UK-based study are also reflected in international studies of practice, which highlight that such adaptations are not unusual [20,21]. Therefore, further investigation of practice would be beneficial. However, SSD and DLD are heterogeneous disorders with any number of different areas being the focus for therapy [4,6]. When exploring aspects of current practice, we might, therefore, consider which areas to focus on, based on age, the current evidence base, and patient and public involvement (PPI)/professional engagement. Although receptive and expressive features of DLD may present within a co-occurring profile, Eadie et al. [8] found that for 4-year-olds with SSD, there was a 36.7% co-occurrence with expressive language needs, in contrast with a 20% co-occurrence with receptive language needs. Furthermore, they found that expressive vocabulary skills at 24 months determined SSD status at 4 years, highlighting a clear link between expressive vocabulary and speech development. In addition to its relationship with SSD features through shared difficulties in underlying phonological representations [22], there is a known impact of limited vocabulary in the early years; for example, low oral vocabulary at school entry is associated with adult literacy, mental health, and employability at 34 years [23]. Due to this link between SSD and expressive vocabulary and known functional impacts, expressive vocabulary was selected as the language focus in this study, alongside speech comprehensibility (within a phonological SSD profile) as the SSD focus. Speech comprehensibility refers to others being able to understand the child’s speech within functional contexts [24]. It is vital for children’s functioning in everyday life, with limited comprehensibility resulting in frustration [25], with downstream consequences for mental health [26]. Improvements in speech comprehensibility are often a direct result of improved speech production accuracy [27]. The impact of limited speech comprehensibility is exacerbated in children with limited words at their disposal, as the few words they might have will be unclear to others, thus compounding pre-existing communication barriers. The focus on expressive vocabulary and speech comprehensibility was supported by pre-project PPI and clinician engagement work, as reported in Section 2.1 of this paper.

### 1.4. Implementation and Complex Interventions

When considering the creation of new, theoretically informed interventions for children with DLD/SSD features, thought must be given to how such interventions can be implemented [3]. This is important. Even if an intervention has a strong supporting theory, if it cannot be implemented by SaLTs within their everyday working contexts it is ineffectual [28]. Paediatric SLT interventions may be deemed ‘complex’ as they are characterised by numerous different interacting components and often involve input on different levels, such as direct work with the SaLT and indirect work through a supported parent [29]. Within the Medical Research Council (MRC) guidance for developing and evaluating complex interventions, there are six core elements in intervention development and evaluation. These include the development of key theory, engaging with stakeholders, and identifying key uncertainties. The current study was primarily concerned with investigating key uncertainties relating to clinical practice when providing a combined SSD/DLD intervention and making the characteristics of current practice explicit. This will enable future researchers to integrate the findings into the development of new interventions for young children with a co-occurring SSD/DLD profile, where speech comprehensibility and expressive vocabulary are targeted areas. The aspects of clinical knowledge (uncertainties) that are most relevant for exploring at a granular level were, therefore, identified and are specified below.

### 1.5. Key Uncertainties

As demonstrated in Table 1, key uncertainties relate to aspects of intervention content, format, and delivery. A brief justification of why these key uncertainties are important in relation to providing a combined intervention is given below.

#### 1.5.1. Content

Phonological awareness is important to literacy development [7], and pre-schoolers with a co-occurring SSD/DLD profile are particularly vulnerable to future literacy needs as they get older [1]. Therefore, aspects of phonological awareness might be targeted within a combined intervention. Research also indicates that the techniques and activities used within interventions for DLD or SSD features can have similar characteristics [12]; these similarities might be utilised within a combined intervention. A final content-related key uncertainty is the targets SaLTs set; these might relate to developmental norms or everyday functional impacts, with both having the potential to result in differing intervention content and delivery.

#### 1.5.2. Format

In their scoping review, Rodgers et al. [3] found that approaches within combined DLD/SSD interventions tended to fall into one of two categories, both with an emerging evidence base. The first approach is the integration of speech/language techniques into the same activity or session. The second approach is the alternation/separation of techniques (e.g., one session focuses on speech, while the next session focuses on language). Ultimately, both the content and delivery of a combined intervention would be informed by which underlying approach is taken: to integrate techniques, or to alternate/separate them.

#### 1.5.3. Delivery

When considering how a combined intervention might be delivered, findings from DLD and SSD studies highlight that this could be performed in a variety of ways, from sole delivery with a SaLT in a clinic to delivery through a supported parent in the child’s home environment [12]. Within these studies, there is also high variability in the reporting of dosage and different dosage types (e.g., the number of sessions and the number of times a technique is used within a session) [12]. Dosage levels have the potential to impact intervention effectiveness [30,31] and, therefore, warrant further exploration regarding use in clinical practice.

### 1.6. Current Study

In summary, to date, the evidence base is limited regarding interventions for young children with co-occurring DLD/SSD features, including those that are feasible for delivery within clinical settings.

To address these gaps in the knowledge base, this paper reports on a survey design study regarding SaLT reports on current practice (content, format, and delivery) for children with DLD/SSD features, where expressive vocabulary and speech comprehensibility are areas of focus. By making such clinical expertise explicit, vital knowledge can be obtained and used to both inform practice and the development of new interventions [29].

The overarching research question was as follows:


*What are the characteristics (i.e., content, format, and delivery) of UK SaLTs’ practice when targeting features of both DLD (expressive vocabulary) and SSD (speech comprehensibility) in pre-schoolers with a co-occurring profile?*


There were also a number of sub-questions, as presented in Table 2 below.

#### Sub-Questions: Framing the Key Uncertainties

The Template for Intervention Description and Replication (TIDieR) [32] is a commonly used framework for intervention reporting in healthcare research, which specifies different intervention components. The dose form framework [33] also specifies intervention components but contains additional detail specific to paediatric SLT. Therefore, both were used to help transform the key uncertainties into sub-questions. An outline of key sub-questions (based on the key uncertainties) and how they relate to the TIDieR and dose form framework is presented in Table 2.

## 2. Materials and Methods

The above research questions were addressed using a quantitative online survey. This was to enable the statistical analysis of technique and intervention use by clinicians from across the UK.

This research was approved by the Language and Communication Science Proportionate Review Committee at City University of London (project number: ETH2223-2120). Survey respondents provided informed consent by completing an online consent form prior to starting the survey.

### 2.1. Patient and Public Involvement (PPI) and Clinician Engagement

PPI and clinician engagement work informed all parts of the research process, including confirming the appropriateness of the selected target areas, survey generation, and data analysis. Further details are given below, in accordance with the Guidance for Reporting Involvement of Patients and the Public (GRIPP 2), short form [34] (Appendix A).

#### 2.1.1. Identifying pSSD (Phonological SSD) and DLD Target Areas (Pre-Study)

The pre-study work included input from a parent advisory group (3 parents of children with DLD/phonological SSD) and 86 UK SaLTs who attended a talk by the lead researcher at two UK Clinical Excellence Network (CEN) meetings, followed by a poll. The target areas of 1. *increasing expressive vocabulary* and 2. *increasing understandability to others in everyday contexts* were identified as appropriate by both SaLTs and parents. The importance of target areas having ‘real world’ meaning for the child, including the ability to make themselves understood to others in daily life, aligns with children’s views in the area [25]. They also align with wider qualitative research with parents, which has shown that parents may be anxious about their child’s ability to communicate in everyday life [35]. The SaLTs’ rationale for targeting expressive vocabulary was that many of the children on their caseload were not at a level where they could work on advanced areas of language, such as syntax or morphology, yet. This is supported by the wider literature, which highlights that children with DLD often have vocabularies that are limited in breadth and depth [36]. Our pre-study PPI and clinician engagement work was important; by utilising evidence as well as engaging relevant individuals, the findings of the current study are more likely to be relevant to the ultimate users of the evidence it will produce [37].

#### 2.1.2. This Study’s Project Steering Group

The project steering group consisted of a parent of a child with DLD/SSD, an adult with DLD, three specialist UK SaLTs (one with specialist equality, diversity, and inclusion expertise), a specialist early-years teacher, and a bilingual/multilingual educational family support worker. The aim of the steering group work was to embed a range of diverse perspectives into the methodology and interpretation of the study findings to enhance the relevance of this research to everyday clinical practice and families receiving it.

In May 2023, each steering group member had a 1:1 meeting with the project lead to discuss the content for the initial survey questionnaire prior to piloting. In June 2023, the impact of these discussions on the survey content was fed back to the group in an online meeting. Impacts included changing some of the question wording (see Section Terminology: Intelligibility and Comprehensibility below), taking out unnecessary questions, and giving an example for each technique mentioned to maximise a shared understanding between survey participants. A further 1:1 meeting with each group member took place in November 2023, where the survey findings and implications were discussed. Key points raised included how to balance functional and developmental targets, challenges in working on phonological awareness when a child may not have the pre-requisite attention and meta-cognitive skills, and the apparent lack of low-tech augmentative and alternative communication (AAC) within SSD research. These points have been integrated into Section 4 of this article. The impact of the steering group work was recorded in an ‘impact log’ as it happened, and excerpts of this can be found in Appendix A.

### 2.2. Survey Content

The survey content was framed around the two target areas of expressive vocabulary and the understandability of the child’s speech to others in everyday life. However, the project steering group decided that content on functional strategies *and* direct work on speech production for the latter target was needed. This is because although environmental strategies will directly impact the child being understood in their contexts, direct work on speech production accuracy will likely also have (longer-term) knock-on effects on the child being understood [27].

#### Techniques, Activities, and Survey Creation

Intervention techniques and activities for the target areas were informed by prior reviews [12,19]. Techniques can be defined as specific teaching behaviours that are suspected to lead to change during an intervention [33]. An example of expressive vocabulary techniques included labelling and expansion, and an example of speech techniques included recasting and using multi-sensory cues. Activity types spanned naturalistic activities, such as everyday routines, and more structured activities, such as picture posting. Content for the survey was refined within the piloting process (described below). The full survey questionnaire can be found in Appendix A.

An initial draft of the survey questions was generated in Qualtrics, based on the key uncertainties stated above. Display and skip logic were used to enhance efficiency and minimise time wastage for participants. Ranking questions were set to present responses in a random order for each participant to minimise order bias. Questions related to a broad description (vignette) of a pre-school child with co-occurring DLD/phonological SSD features, where expressive vocabulary and speech comprehensibility were the joint areas of focus. The vignette was kept deliberately broad to capture patterns in clinical practice based on the core clinical profile alone. Open questions requiring free-text responses were kept to a minimum to reduce the time burden for participants. Questions were grouped and then provided in the following sequence: 1—intervention targets, 2—expressive vocabulary techniques, 3—speech techniques, and 4—technique delivery. A small number of open questions were provided to give SaLTs the opportunity to expand their responses.

### 2.3. Survey Piloting

Prior to going live, the survey was piloted in four ‘think-aloud’ interviews [38], with SaLTs from three different NHS trusts in the UK. The process involved using comprehension probes such as “did—make sense to you?” These were asked when the SaLT was at the end of each section of the survey to minimise interruptions to their experience of completing the survey, whilst the information was still fresh in their minds. Changes made to the survey following the think-aloud interviews included minor amendments to the vignette description and reminders at the start of each section of the survey of the clinical profile of the vignette. An example of the recording of the think-aloud interviews and subsequent changes made is given in Appendix A.

#### Terminology: Intelligibility and Comprehensibility

A matter of contention both within the steering group discussion and the piloting process was how best to term *‘the child being understandable to those around them in everyday contexts’*. In their recent Delphi study, Pommee et al. [24] defined speech intelligibility as a listener’s ability to accurately recognise and decode individual words from an utterance, and comprehensibility as the listener’s ability to understand the overall meaning of the utterance within its broader context. During the steering group discussion and the piloting process, it was raised that comprehensibility is not a term readily used or understood by UK SaLTs, who would still use intelligibility to describe a child’s speech being understood within their environmental context. This interchangeable use of terms is also reflected in the literature, where studies describing speech comprehensibility use the term intelligibility [39]. For this reason, although ‘comprehensibility’ will be used in this paper, we used the term ‘intelligibility’ in the survey itself to support SaLTs’ understanding.

### 2.4. Inclusion Criteria and Recruitment

Participants were required to be HCPC SaLTs registered within the UK, with at least 1 year’s experience of working in this clinical area within the past two years. Retired clinicians and those who had not worked in this clinical area for over two years were also excluded. It was decided that the survey would include UK-based SaLTs only due to international variation in service structures and understanding of key terminology.

A sample size of 100 full responses was aimed for; similar sample sizes are observed within other surveys of clinical practice within paediatric SLT [40]. As children presenting with features of DLD and SSD form a core part of a generalist early years’ caseload [41], this was deemed achievable, subject to a successful recruitment strategy.

The survey (Appendix A) was advertised via the research team’s networks on Twitter/X via email to 11 relevant UK Clinical Excellence Networks (CENs) and on the RCSLT Research Champions online forum. The survey was open from 25 July to 13 October 2023. In total, there were 119 responses: 108 full and 11 partial. A partial response was defined as one where a respondent had answered any questions beyond the initial screening and demographic questions but had not reached the end of the survey.

### 2.5. Analysis

To prevent the potential for multiple entries from the same individual, survey metadata were used to confirm that all respondent IP addresses were different. The data were then imported from Qualtrics onto SPSS version 29.0.2.0 (20). The data were analysed using descriptive statistics, and mean ranks enabled the researchers to demonstrate the average mean responses across the group, including relative preferences within ranking questions. Inferential statistics were not performed due to the sample size.

## 3. Results

### 3.1. Demographics

There were 119 responses, of which 108 were complete and 11 were partial. For transparency, raw data are presented alongside any percentages given due to the participant numbers declining towards the end of the survey. Of the 119 respondents, 95% (113/119) were female, 2.5% (3/119) were male, and 2.5% (3/119) preferred not to state. Most respondents worked for the UK National Health Service (NHS) only (102/119 = 85.7%), or within both the NHS and independent practice (5/119 = 4.2%). The top three settings in which they provide their service were clinic only (34/119 = 28.6%), within both clinic and educational settings (29/119 = 24.4%), and in educational settings only (24/119 = 20.2%). Years of experience and geographical location details are given in Table 3 below.

### 3.2. Intervention Content and Format

#### 3.2.1. Targets

For both expressive vocabulary and speech comprehensibility, targets relating to functional outcomes (i.e., sounds having the most impact on the child being understood by others and vocabulary of importance to the child’s everyday life) were prioritised. This is reflected in the mean ranks of 1.32 and 1.97, respectively (Figure 2 and Figure 3), where a lower mean rank indicated a stronger preference. Conversely, targets based on typically developing norms for both expressive vocabulary and speech comprehensibility had the highest mean ranks, reflecting that they were the least preferred target types overall (2.98 and 3.08, respectively).

There was a strong preference for both expressive vocabulary and speech comprehensibility targets to be integrated into the same activities or therapy sessions, as demonstrated in Table 4.

#### 3.2.2. Phonological Awareness

Most respondents (116/119 = 97.5%) reported that they would also target phonological awareness within their intervention, in addition to expressive vocabulary and speech comprehensibility. Of the respondents who said they would focus on phonological awareness, the most frequently stated phonological awareness activities related to syllable segmentation (96/116 = 82.3%).

#### 3.2.3. Expressive Vocabulary Techniques

The selection of expressive vocabulary techniques ranged from the cloze procedure, which was the least popular (74/115 = 64.3%), to labelling, which was the most popular (115/115 = 100%). After labelling, the most frequently selected techniques included choices (112/115 = 97.4%), expansion (108/115 = 93.9%), and focused auditory stimulation (104/115 = 90.4%).

Overall, 69/115 (60%) of respondents stated that they would prioritise some of their selected techniques over the others, with labelling being the most prioritised technique (mean rank: 1.9), followed by focused auditory stimulation (2.93), expansion (3.05), and then choices (3.89).

A key justification for the technique selection included ease of implementation, with respondents commenting on how techniques could be easily used by adults in the child’s everyday environment:


*“Labelling and expansion can be built into motivating play activities and replicated by parents.”*
(respondent 1)

A further justification for the expressive vocabulary technique choice was the perceived importance of exposure to/frequent repetition of vocabulary:


*“Clinical experience of input supporting output changes.”*
(respondent 2)

#### 3.2.4. Speech Comprehensibility Techniques

The selection of speech techniques ranged from discriminating between non-speech sounds, which was the least popular (36/111 = 32.4%), to the most popular technique of speech recasts (111/111 = 100%). The joint-second most popular techniques were discriminating between word pairs that differ by a key sound and using multi-sensory cues (both 109/111 = 98.2%). They were closely followed by auditory bombardment (99/111 = 89.2%) and then broad target recasts (97/111 = 87.4%).

Overall, 64.9% (72/111) of respondents stated that they would prioritise some of their selected speech techniques over the others. However, in contrast with the expressive vocabulary techniques, there was less differentiation between the top four prioritised techniques for speech comprehensibility. Identifying a word based on sound contrasts (2.43), auditory bombardment (2.62), using multi-sensory cues (2.78), and speech recasts (2.96) were the most prioritised, followed by discrimination between non-speech sounds (4.00) and broad target recasts (4.36). This could be due to the closely interconnected nature of the speech techniques, for example,


*“Discrimination activities also create opportunities for auditory bombardment and recasting, and support vocab too.”*
(respondent 3)

A common justification for the technique selection was related to the importance of exposure, which might be achieved using any one of the input-related techniques:


*“My approach is based on input modelling—I am trying to highlight key sounds to the child, so that over time they can add them to their inventory.”*
(respondent 4)

When asked whether techniques to elicit speech output from the child were deemed essential, 72/108 (66.7%) of respondents stated yes. However, free-text comments indicated overlaps in the underpinning rationales behind both yes and no responses to this question, with the child’s “readiness” being a key consideration:


*“I think unless the child is very aware of their difficulties, or is very shy, it is important to have a go at output.”*
(respondent who ticked yes)


*“Depends on attention, awareness, and self-confidence.”*
(respondent who ticked no)

#### 3.2.5. Environmental Strategies for Comprehensibility

In addition to speech techniques for improving speech sound production, respondents were unanimous in their incorporation of strategies for everyday comprehensibility into their intervention. The top three environmental strategies selected included the adult asking the child to “show me” (111/111 = 100%), encouraging the child to gesture/act out what they are trying to say (110/111 = 99.1%), and making pictures available to refer to (109/111 = 98.2%).

#### 3.2.6. Activities

Respondents would use a variety of different activity types in which to incorporate their expressive vocabulary and speech techniques (Figure 4). Electronic activities (e.g., app games) were the least preferred activity type for both, selected by 25/114 (21.9%) of respondents for expressive vocabulary, and 36/108 (33.3%) of respondents for speech. Preferences for play activities for both speech and language followed similar patterns (see Figure 3). However, there was still some contrast in the most preferred activity types. For expressive vocabulary, the most preferred activity types tended to be more naturalistic in focus, and included routines (109/114 = 95.6%), child-led play (108/114 = 94.7%), and storybooks (106/114 = 93%). In contrast, for speech, the most preferred activity types were posting pictures and turn-taking games (both at 103/108 = 95.4%), followed jointly by ‘what’s in the bag?’ and picture matching (both at 96/108 = 88.9%).

### 3.3. Delivery

Respondents expressed a preference for holistic delivery of intervention techniques across locations and not limited to just the SaLT as deliverer (Figure 5 and Figure 6). The most popular strategies for supporting significant others in delivering techniques included the SaLT modelling the technique to them (108/108 = 100%), written guidance to take away (106/108 = 98.1%), discussion to identify the best time for parents to deliver the technique (105/108 = 97.2%), and providing picture prompts (97/108 = 89.8%). The least popular methods of support were videoing the parent performing the technique to use as a basis for discussion (75/108 = 69.4%) and showing the parent a video of someone else performing the technique (69/108 = 64%).

#### Dosage

When asked, 82.4% (89/108) of respondents reported that they would consider dosage within their intervention. The joint most popular aspects of monitoring and planning the dosage to be delivered were the total number of sessions/teaching episodes and number of times a technique was used per session/teaching episode (both 69/89= 77.5%), followed by the number of times a technique was used within a single therapy activity (65/89 = 73%) (Figure 7).

Free-text comments highlighted that a common underpinning rationale for dosage choice was the evidence base. However, respondents also said that dosage was dependent on the child’s response in therapy, and that it was not always practical to record dosage whilst conducting a therapy session:


*“Very dependent on the child and might change constantly—I just get in as much as possible.”*
(respondent 5)


*“Sometimes it’s difficult to count number of repetitions over the course of a session/intervention. I aim for as many repetitions as possible, but this is not always numbered.”*
(respondent 6)

## 4. Discussion

This study addressed key uncertainties relating to the reported content, format, and delivery characteristics of UK SaLTs’ intervention when targeting both expressive vocabulary and speech comprehensibility in a combined intervention. The findings highlight a preference for integrating the expressive vocabulary and speech targets within a holistically delivered intervention, involving direct work with the SaLT in the clinic as well as with a supported adult in the child’s everyday environment. The activity preferences for both expressive vocabulary and speech were similar, although there was a greater preference for structured activities for speech. Further discussion regarding pertinent findings is given below.

### 4.1. Functional and Developmental Focus

The prioritisation of functional targets over those relating to developmental norms has been evidenced in recent investigations of clinical practice in paediatric SLT [42]. Targets with direct relevance to the child’s participation in everyday life are likely to have a direct positive impact on emotional wellbeing and relationships [25]. However, this preference for functional targets is not conclusive across all contexts; for example, a study by Farquharson et al. [43] found an over-reliance on speech norms by SaLTs in US schools. It can be argued that targets based on developmental norms are not completely independent of functional impact, with improvements in speech and language capability likely to have an indirect impact on the child’s everyday functioning further down the line. Additional developmental (linguistic) considerations also apply to children with a co-occurring phonological SSD/DLD profile. For example, phonological representations can be strengthened via the acquisition of expressive vocabulary [22], which emerge in response to adults using language facilitation techniques (e.g., language modelling, as identified in this study). Through the gradual acquisition of phoneme sequences within their newly acquired vocabulary [44], longer-term speech production might be impacted, increasing the child’s comprehensibility. However, this process is unlikely to happen instantaneously. When developing future interventions for this group, researchers might consider how best to balance immediate, positive functional impacts with developmental norms, including the strengthening of underlying phonological representations. It is encouraging to see a growing exploration of measuring both developmental (impairment-focused) and functional outcomes in the literature, for example, Cunningham et al. [45], who conducted an evaluation of both participation and impairment outcomes using the Hanen target word approach.

Within SSD, the pipeline from ‘improved production’ to ‘everyday impact’ is conceptualised by Baker et al. [27], who describe eight domains of outcome measurement, ranging from the most proximal domain 1 (e.g., the production of specific target sounds) to the most distal domain 8 (the impact of the child’s SSD on other people in the child’s life). Speech comprehensibility sits midway, in domain 5, with it typically following an improvement in direct speech production. Of note within the current study, SaLTs were unanimous in their use of environmental strategies for improving comprehensibility, which included paper-based aided and alternative communication (AAC), such as picture boards. Although not a common feature of published SSD interventions, the use of paper-based AAC and other environmental strategies has the potential to support the child’s comprehensibility in the short term, whilst their speech production is still developing [46]. Therefore, when developing new interventions for this group, this hybrid approach of a) improving production (through direct speech work or sound awareness) and b) functional strategies (such as paper-based AAC) may be warranted to improve the child’s life as soon as possible, whilst their speech accuracy is still developing.

A further functional consideration arising from the results is the child’s ‘readiness’ for intervention, and how this might influence the SaLT’s decision to directly elicit output from them. This finding aligns with the wider qualitative literature, highlighting that child temperament and personality can influence the intervention strategies selected [20,47]. A different but related matter is the child’s attention levels and how this might influence the intervention strategies chosen and the child’s response. This is a particularly important consideration for children with co-occurring SSD/DLD features, who have been found to present with significantly increased ADHD symptoms compared with their peers [48]. It is evidenced that language develops within the context of back-and-forth, meaningful interactions. As demonstrated by Romeo et al. [49], the language pathways of the brain become activated when children are engaged in motivated and meaningful interactions, rather than passively listening. Such evidence provides support for SaLTs’ reported use of language enrichment techniques, such as modelling and expansion, which can be incorporated into a variety of different activities depending on what the child is most motivated by (and, therefore, more likely to attend to).

### 4.2. Integrating Speech and Language

Although within the literature targets for speech and language might be delivered separately within a combined intervention (e.g., alternating one session for speech with one for language) [3], findings highlighted that for children in this age group, and for these target areas, SaLTs have a clear preference for integrating speech and language techniques into a single session/episode instead. Prior research has indicated that integrating speech and language techniques into a single session/episode can be effective; however, the same research also showed that alternating between one session for speech, followed by one session for language, was even more effective [14]. Crucially, this research targeted different outcomes (morphosyntax rather than expressive vocabulary), and, therefore, it is not possible to compare like-for-like. We might look to evidence from other paediatric client groups for early indicative support for an integrated approach (where expressive vocabulary and speech are concurrently targeted), such as ‘enhanced milieu teaching within phonological emphasis’ in children with cleft lip and palate [50]. Although findings highlight the potential for the integration of techniques, thought needs to be given to the operationalisation of this. The survey findings highlighted that although there is some overlap in activities selected for speech and language, preferred speech activities still tended to be more structured. This is in keeping with previous research findings [12]. Therefore, speech and language techniques might be best integrated at a whole-session level (rather than individual activities), where language techniques within more naturalistic activities are alternated with speech activities in more structured activities.

### 4.3. Intervention Techniques (Input and Output)

For both expressive vocabulary and speech, the child’s practice of target items (output) is known to facilitate their learning process [30,31]. However, as previously mentioned, the survey findings raise an interesting concern regarding those children seen within clinical practice who may not be ready for this direct work. With younger children in the pre-school years, although providing output practice is optimal, progress may still be made using input-related techniques, such as language modelling [11] and sound exposure [9]. Future interventions might, therefore, consider how to incorporate flexibility within the intervention process to suit the personality needs of different children who might be at different stages in their readiness for direct production work. Such flexibility is not unusual within complex interventions, where emphasis is based on real-life implementation [29].

### 4.4. Phonological Awareness

Given the shared linguistic underpinnings in DLD/SSD [4], it is perhaps unsurprising that the majority of respondents reported that they would target phonological awareness (PA) in addition to language and speech output within their interventions (with syllable segmentation being the most reported PA area to target). Difficulties with the production of polysyllabic words appear to be a persisting feature for children with a history of expressive language needs [51] and may take place in the context of wider cross-domain difficulties with sequential learning in children with DLD features [52]. Such findings indicate potential short- and long-term benefits of specifically targeting syllable segmentation in young children. However, this relative benefit will likely be child-dependent; for example, children who already demonstrate good syllable segmentation skills may instead benefit from phoneme-level tasks (e.g., initial sound identification) [53]. A focus on phonological awareness within phonological SSDs is in line with national [54] and international [55] guidelines, although specific guidelines for *co-occurring* SSD/DLD, and, therefore, the targeting of phonological awareness within a co-occurring profile, are lacking. Although more research is warranted in this area, SaLTs might consider using assessments, such as the Newcastle Assessment of Phonological Awareness (NAPA) [56], to guide their setting of PA targets.

### 4.5. Dosage Considerations

Previous research has highlighted a dearth of dosage-based information within intervention studies [12,30], despite adequate dosage being central to the success of both SSD and DLD interventions [30,31]. The findings highlight that the provision of dosage-related information would be helpful to SaLTs in their everyday practice, providing further support for dosage specifications to be integrated into intervention descriptions. However, the variation in therapy responses, even in children with the same clinical profile, remains a challenge to applying specific dosage specifications within everyday clinical practice. At the service level, challenges may exist regarding what dosage SaLTs are funded to provide within a fragmented NHS healthcare system [57]; such service constraints are also evidenced internationally [58]. Dosage complexities might be carefully navigated by researchers considering how dosage might be made flexible, using development-evaluation trials to aid dosage refinement [29]. Examples of such intervention dosage refinements are starting to emerge within the field; for example, a study by Zulkifli et al. [59] found that for the Hanen “It Takes Two to Talk” programme, children with fewer risk factors still could benefit from a lower dosage when compared with those with higher risk factors.

### 4.6. Limitations

When considering the study findings, it should be acknowledged that survey respondents were likely to have a pre-existing interest in this clinical area, which is why they took the time to complete the survey. Therefore, the findings may not be fully representative of all UK SaLTs who come into contact with this client group. Additionally, surveys cannot always pick up on the subtleties in clinical practice or opinion. A related consideration is that reported practice does not necessarily equate to effective practice. Future work needs to build on these findings by exploring the nuances of practice in more detail and the effectiveness of reported practice within exploratory intervention trialling (where components of reported practice are included within the intervention being trialled).

Due to the sample size of the survey, it was not appropriate to perform subgroup analysis based on geographical location. A similar consideration is that the vignette was kept deliberately broad (as appropriate for the purpose of the survey); however, SaLT intervention will likely be influenced by individual factors in addition to the child’s clinical profile. Future research could explore the impact of geographical location (specifically, service delivery structure within services in different locations) and the specific impact of individual factors on intervention planning for this client group.

The study findings need to be viewed in the context of intervention for two specific areas (expressive vocabulary and speech comprehensibility) and, therefore, may not reflect clinical practice where different outcomes are targeted. Although the direct generalisation of the results to interventions focusing on alternative outcomes is cautioned, we may still consider how the findings could inform the wider sphere of paediatric SLT intervention. For example, this study raised interesting questions regarding the role of dosage, functional vs. norms-based therapy, and the role of the child’s personality/‘readiness’ within the intervention process. These topics are unlikely to be unique to this client group alone. Therefore, there may be value in further exploration of these topics when considering intervention for other under-researched paediatric client groups.

### 4.7. Directions for Future Research

As mentioned within the limitations, future research enabling nuanced analysis of service delivery and individual child/family factors would be beneficial. Additionally, further exploration of intervention content related to phonological awareness and AAC/functional comprehensibility strategy use is needed. For example, what AAC/functional comprehensibility strategies are used could be influenced by service delivery factors. Phonological awareness was not explored in depth within the survey, yet responses indicated that SaLTs find phonological awareness just as important to target as (direct) speech and language within this client group. Future research is needed to probe the role and effectiveness of incorporating phonological awareness into interventions for this group further.

### 4.8. Summary

This study made explicit the reported practice of UK SaLTs when implementing an intervention for pre-school children with co-occurring features of DLD and phonological SSD, where expressive vocabulary and speech comprehensibility were targeted. These findings can be used to inform the development of new interventions for this under-researched group. There was some alignment with SaLT practice and the current evidence base, although discrepancies were also evident, e.g., SaLTs’ use of AAC and environmental/functional strategies for everyday life. The findings provide considerations relating to interventions that may also have relevance to the wider paediatric SLT community, including SaLTs’ views on the importance of dosage and balancing functional impact with developmental norms.

## Figures and Tables

**Figure 1 children-12-01568-f001:**
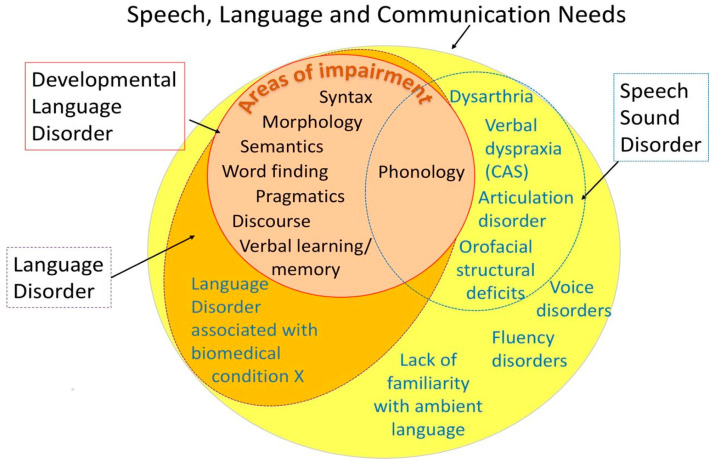
Venn diagram of speech, language, and communication needs [4].

**Figure 2 children-12-01568-f002:**
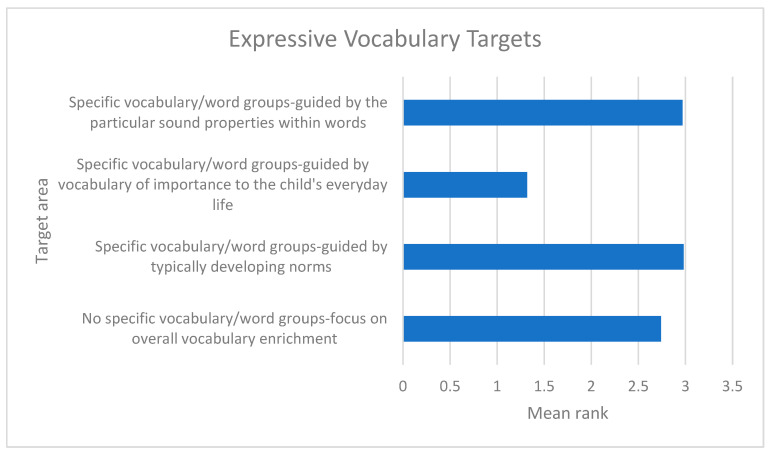
Mean ranks for expressive vocabulary targets (lower mean rank = more preferred).

**Figure 3 children-12-01568-f003:**
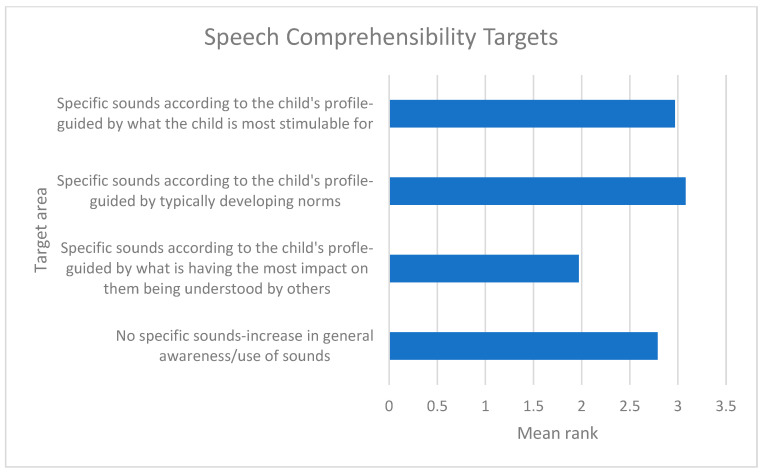
Mean ranks for speech comprehensibility targets (lower mean rank = more preferred).

**Figure 4 children-12-01568-f004:**
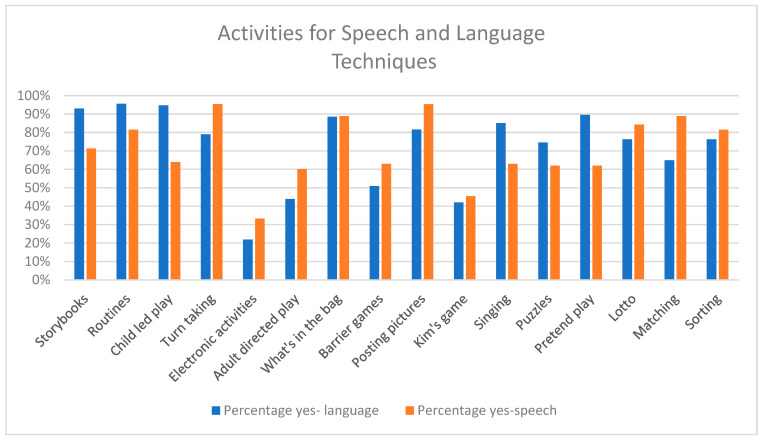
Types of activity for incorporating techniques.

**Figure 5 children-12-01568-f005:**
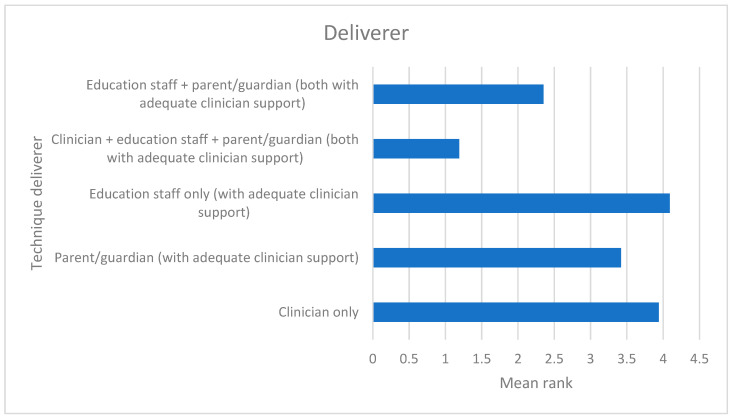
Preferred deliverer (lower mean rank = more preferred).

**Figure 6 children-12-01568-f006:**
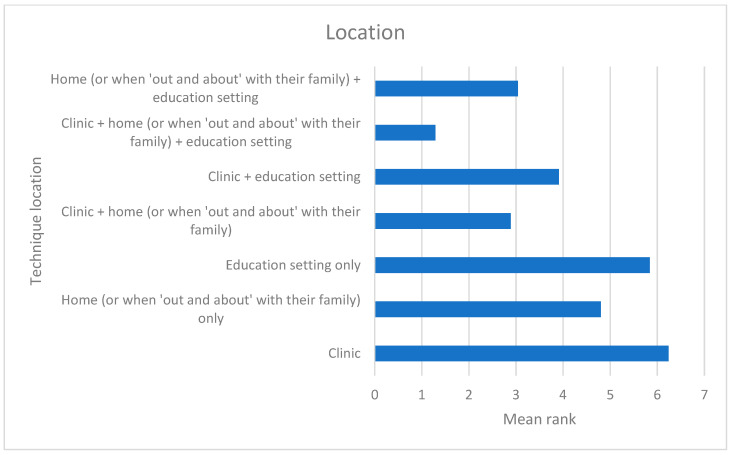
Preferred location for delivery (lower mean rank = more preferred).

**Figure 7 children-12-01568-f007:**
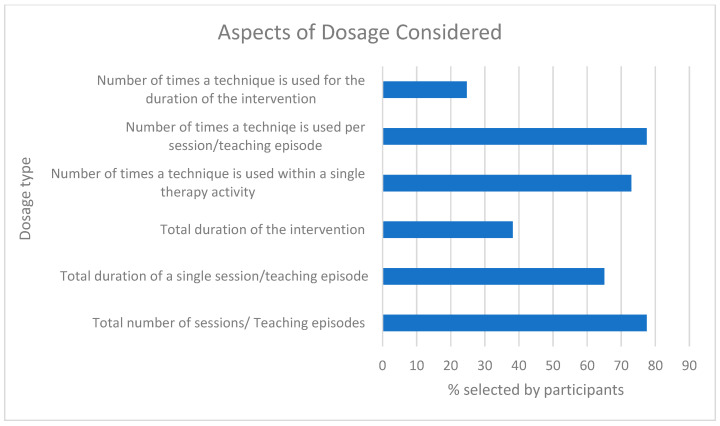
Aspects of dosage considered.

**Table 1 children-12-01568-t001:** Key uncertainties relating to content, format and delivery SaLT’s reported practice when providing a combined intervention for co-occurring pSSD/DLD features.

Intervention Content	Intervention Format	Intervention Delivery
Techniques and activities SaLTs would use within a combined intervention, including environmental strategies.	How SaLTs would combine speech/language targets within their combined intervention.	Who the SaLTs would choose to deliver their combined intervention, and how non-SaLTs are supported if they are delivering it.
Whether SaLTs would incorporate phonological awareness work into their combined intervention.		Where the SaLT would deliver their combined intervention.
How SaLTs would characterise their intervention targets within a combined intervention.		Whether dosage is important to SaLTs when delivering their combined intervention (e.g., *aiming for a specific number of sessions or number of times a technique is used within an activity*).

**Table 2 children-12-01568-t002:** Mapping of sub-questions (key uncertainties) onto intervention frameworks.

Sub-Questions (Key Uncertainties) for Exploration	TIDieR [32]	Dose Form Framework [33]
What types of target characteristics (i.e., functional or impairment-based) are prioritised, and in what format are they most likely to be delivered?	TIDieR criteria 4—what?	Method of instruction; procedure
What intervention techniques are SaLTs most likely to implement? Are they hierarchical or of equal importance?	TIDieR criteria 4—what?	Techniques
What activities would SaLTs incorporate their selected techniques into?	TIDieR criteria 4—what?	Intervention contexts: activities
Who would SaLTs recommend deliver these techniques, where, and what aspects of dosage might they consider/not consider?	TIDieR criteria 5—who? TIDieR criteria 7—where? TIDieR criteria 8—when/how much?	Method of instruction; procedure
If the deliverer is not the SaLT, how would others be supported to deliver these techniques?	TIDieR criteria 4—what?	Method of instruction
Would SaLTs also incorporate phonological awareness work into their intervention?	TIDieR criteria 4—what?	N/A
What environmental strategies (if any) would SaLTs recommend?	TIDieR criteria 4—what?	N/A

**Table 3 children-12-01568-t003:** Experience and geographical location of respondents.

Years of Experience	N and Percentage of Respondents
Under 3 years	(16/119) 13.4%
3–10 years	(40/119) 33.6%
11–15 years	(22/119) 18.5%
16–20 years	(13/119) 10.9%
21+ years	(28/119) 23.5%
**Geographical Location**	
East of England	(13/119) 10.9%
East Midlands	(4/119) 3.4%
London	(14/119) 11.8%
Northern Ireland	(11/119) 9.2%
Northeast England	(6/119) 5%
Northwest England	(7/119) 5.9%
Scotland	(6/119) 5%
Southeast England	(22/119) 18.5%
Southwest England	(15/119) 12.6%
Wales	(6/119) 5%
West Midlands	(12/119) 10.1%
Yorkshire and the Humber	(3/119) 2.5%

**Table 4 children-12-01568-t004:** Formatting of targets.

Target Format	Mean Rank (Descending Order of Preference)
Integration of speech and vocabulary targets into the same activities within a therapy session (i.e., *a single therapy activity includes content to develop both speech and language*)	1.59
Alternation between one activity targeting speech and one activity targeting vocabulary within a therapy session	1.99
Alternation between a session for speech targets and a session for vocabulary targets for the duration of the intervention	3.58
Target vocabulary in the first half of the intervention and speech in the latter (i.e., *activities in the first half of the intervention include content just for language; activities in the second half of the intervention include content just for speech*)	3.71
Target speech in the first half of the intervention and vocabulary in the latter (i.e., *activities in the first half of the intervention include content just for speech; activities in the second half of the intervention include content just for language*)	4.13

## Data Availability

The original data presented in this study are openly available in FigShare at https://doi.org/10.25383/city.27959865.v1 (URL accessed 17 November 2025).

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
