# Peer review of "Concurrent Targeting of Expressive Vocabulary and Speech Comprehensibility in Pre-Schoolers with Developmental Language Disorder and Phonological Speech Sound Disorder Features: A Survey of UK Practice"

_children, 2025, doi:10.3390/children12111568_

Round 1

Reviewer 1 Report

Comments and Suggestions for Authors

See attachment.

Reviewer 2 Report

Comments and Suggestions for Authors

 Thank you very much for your invitation to review the manuscript with the title “Concurrent targeting of expressive vocabulary and speech comprehensibility in pre-schoolers with co-occurring features of developmental language disorder and consistent phonological speech sound disorder.”

The manuscript has much potential to be a useful and valuable tool for the field. Furthermore, this study has the scope to clarify the content and format of such interventions when preschoolers have co-occurring disorders like DLD and SSD. The authors addressed the main question. The topic is relevant and has novelty as there is little literature to discuss this disorder together, and this will help those professionals who work on these populations, especially SLPs in children. As this manuscript did not add any new intervention approach, at least the new addition is on the empirical description of the therapists. This need to be more clear by the authors in their manuscript. Below, I have a few more suggestions for the authors.

The title is very long (30 words around) and there is a need to be shortened if it possible. The authors should keep some keywords in the title.

The abstract reflects the work that has been done, but the methods need more details.

Introduction

This section is very descriptive and very informative. The authors have a flow in their writing, and the text follows good academic writing. They follow a subsection model that helps better understand the context in depth.

-The first paragraph has many statements but lacks references.

Methods
This section is quite descriptive, and this will allow others to reproduce such research. They also describe the whole process and the pilot. One more thing is that the authors mentioned the ethical approval. They explain how they handle the terms intelligibility and comprehensibility.

They also exaplain how did they find the sample, and how exactly did they call them to participate. One comment is that the sample would be characterized as a sampling bias due to their self-selection.

Another thing that would influence the results is that there is no info about the children’s background as these details would impact the intervention planning.

The 2.4 subsection referred to the criteria and the recruitment.
For the analysis they used descriptive statistics in SPSS.

Results

The authors included 4 tables and 7 figures to present their results. These have sufficient quality and clear. The figures are helpful as they present mean ranks. The description in text is in consistency with the tables and figures. The findings answered the scope of the study.

Discussion

This section has 7 subsections. Every subsection delivered a scope, to discuss and source the key findings.

- There is a need about connect the theory in language/phonological development.
- They reported functional intervention and I think a connection with socioparagmatic language development would be beneficial.

The conclusions reflects the data of the study, but as the study didn’t evaluate any outcomes the authors should carefully refer to the tone of this study as a survey. A revision to that scope is required.

References

The references are appropriate, relevant and are connected with the topic. 

Round 2

Reviewer 1 Report

Comments and Suggestions for Authors

Overall, the authors have made meaningful revisions and provided thoughtful responses to prior comments. The manuscript has improved in clarity, organization, and focus. However, several points would still benefit from further attention to enhance transparency and interpretability.

1. Clarification on the role of phonological awareness

In response to the earlier comment regarding the role of phonological awareness, I appreciate the authors’ clarification and discussion of its relevance. However, if phonological awareness is indeed a frequently targeted skill in clinical practice but was not examined in the current survey, this omission should be explicitly acknowledged as a limitation. It would also strengthen the paper to include a brief discussion of future research directions, particularly regarding how phonological awareness could be systematically investigated in subsequent studies.

2. Representativeness and potential for subgroup analyses

Regarding representativeness and sample size, the authors have provided additional context, which is helpful. However, it remains unclear whether any demographic or service-related variables (e.g., region, service setting, client demographics, language background, functioning level, or AAC use) were collected and could potentially support subgroup analyses. If such data exist, a brief exploration or descriptive summary would add value. If subgroup analyses are not feasible due to sample size or data constraints, this should be explicitly stated as a limitation and identified as a future research direction, rather than being omitted.

3. Consideration of AAC

If relevant data are available, a subgroup analysis or descriptive summary would be informative. If such analysis is not feasible, this should be explicitly acknowledged as a limitation and discussed as a future research direction.
